# ApoE Lipidation as a Therapeutic Target in Alzheimer’s Disease

**DOI:** 10.3390/ijms21176336

**Published:** 2020-09-01

**Authors:** Maria Fe Lanfranco, Christi Anne Ng, G. William Rebeck

**Affiliations:** Department of Neuroscience, Georgetown University Medical Center, 3970 Reservoir Road NW, Washington, DC 20057, USA; ml1358@georgetown.edu (M.F.L.); cn472@georgetown.edu (C.A.N.)

**Keywords:** apolipoprotein E, cholesterol, lipid homeostasis, neurodegeneration

## Abstract

Apolipoprotein E (*APOE*) is the major cholesterol carrier in the brain, affecting various normal cellular processes including neuronal growth, repair and remodeling of membranes, synaptogenesis, clearance and degradation of amyloid β (Aβ) and neuroinflammation. In humans, the *APOE* gene has three common allelic variants, termed E2, E3, and E4. *APOE4* is considered the strongest genetic risk factor for Alzheimer’s disease (AD), whereas *APOE2* is neuroprotective. To perform its normal functions, apoE must be secreted and properly lipidated, a process influenced by the structural differences associated with apoE isoforms. Here we highlight the importance of lipidated apoE as well as the *APOE*-lipidation targeted therapeutic approaches that have the potential to correct or prevent neurodegeneration. Many of these approaches have been validated using diverse cellular and animal models. Overall, there is great potential to improve the lipidated state of apoE with the goal of ameliorating *APOE*-associated central nervous system impairments.

## 1. Scope of This Review

In this review, we will first consider the role of apolipoprotein E (apoE) in lipid homeostasis in the central nervous system (CNS). We will then summarize data on the effects of *APOE* genotype on apoE function and lipid trafficking in normal functions and in pathogenesis. Finally, we will speculate on ways to increase lipidation of apoE with the goal to decrease apoE-associated CNS impairments. 

## 2. Introduction

ApoE, a 34 kDa protein composed of 299 amino acids, is a member of the superfamily of amphiphilic exchangeable apolipoproteins. It is expressed in hepatocytes, monocytes/macrophages, adipocytes, astrocytes, and kidney cells. In humans, the *APOE* gene has three common alleles, ε2, ε3, and ε4 with frequencies in the US population of: ε2, 7%; ε3, 79%; and ε4, 14%, but this distribution varies widely with ethnicity and geography [1,2,3,4]. 

*APOE4* is considered the strongest genetic risk factor for the development of Alzheimer’s disease (AD). *APOE4* also affects other neurological disorders, including poor clinical outcomes after traumatic brain injury or stroke [5,6], frontotemporal dementia [7,8], Down syndrome [9], and Lewy body disease [10]. About 25% of the people in the U.S. are *APOE4* carriers, and *APOE4* is present in 65–80% of all AD patients. The presence of *APOE4* increases the risk of developing AD by 4-fold (one allele) to 14-fold (two alleles) compared with *APOE3* homozygotes. *APOE2* is considered neuroprotective, albeit with an increased risk of type III hyperlipoproteinemia [11,12]. The allelic combinations give rise to three homozygous (*APOE2/2*, *APOE3/3,* and *APOE4/4*) and three heterozygous (*APOE2/3*, *APOE2/4,* and *APOE3/4*) genotypes. Single point substitutions at amino acid residues 112 and 158 distinguish the three main apoE isoforms: apoE2 (Cys-112, Cys-158), apoE3 (Cys-112, Arg-158), and apoE4 (Arg-112, Arg-158) [13,14]. These substitutions result in structural differences that confer differential receptor and lipid binding abilities, and provide grounds to explain the physiological role of apoE isoforms in AD.

## 3. Structures and Functions of apoE Isoforms

ApoE is a glycoprotein that has two independently folded structural domains separated by an unstructured hinge region. The N-terminal domain (residues 1–191), which forms a four α-helix bundle in which non-polar residues face the inside of the protein, harbors the sequences for binding to the members of the low-density lipoprotein (LDL) receptor family [15]. The C-terminal domain (residues 206–299), that harbors the major lipid binding region, presents a more relaxed structure with α-helices generating a largely exposed hydrophobic surface [15]. The unstructured hinge regions provide a large degree of mobility, which is necessary for the protein to fulfill its primary function [15].

ApoE undergoes a profound conformational change upon lipid binding to accommodate and stabilize the lipids through its amphiphilic α-helices [16]. Following lipid binding, apoE adopts a biologically active conformation that is necessary for its recognition and binding to the members of the LDL receptor family and its internalization [16]. The use of discoidal phospholipid-apoE particles or “nanodisks” have allowed a better understanding of the structure of lipid-bound apoE on some high density particles [16,17]. Lipid binding reorients the α-helices of the C-terminal domain of apoE perpendicularly to the acyl chains of the lipids, circumscribing the edge of the nanodisks [17,18,19]. However, the conformational change on the N-terminal domain has not converged towards a single model.

Calorimetry studies support a model in which the four-helix bundle opens leading to an extended conformation of apoE that wraps around the lipid bilayer of the disc [20,21]. By contrast, low resolution X-ray density and electron paramagnetic resonance favor a model in which apoE folds in a hairpin structure [22]. More recently, the combination of experimental techniques in molecular modeling and simulation support a model in which an open and compact hairpin conformation co-exist in a dynamic equilibrium, and the presence of receptor binding shifts the equilibrium towards the open hairpin conformation [17].

When analyzing the three apoE isoforms independently, X-ray crystallography measurements indicate that the Cys-158 at the N-terminal domain of apoE2 prevents receptor binding by changing the conformation of the lipid side chains residues 136–150 [12,13,23,24]. In apoE3, a salt bridge between Arg-158 and Asp–154 is formed, whereas in apoE4 it is disrupted. In apoE4, Asp-154 interacts with Arg-150, altering the receptor binding region. As a consequence of this new interaction, the side chain of Arg-61 is now in close proximity to Glu-255 in the C-terminal domain, allowing an ionic interaction between these two residues [12,13,23,24]. This interaction is less likely to happen in apoE2 and apoE3, because in these isoforms a structure is favored in which the Arg-61 is internal to the helical domain of the N-terminus [12,13,23,24]. A summary of the structural differences among apoE isoforms is depicted in Table 1.

## 4. CNS apoE Protein

In the CNS, apoE is synthesized in situ primarily by astrocytes, but also by microglia, oligodendrocytes, and to lesser extent injured neurons [25,26,27,28,29]. An additional source of apoE comes from the choroid plexus where apoE is rapidly delivered into the cerebrospinal fluid (CSF) and the brain via the glymphatic fluid transport, in an apoE isoform-dependent manner (apoE2 > apoE3 > apoE4) [30]. Peripheral apoE does not cross the blood–brain barrier (BBB), resulting in two separate pools of apoE, one in the periphery and one in the CNS.

The posttranslational modifications of apoE are different in the periphery and CNS [31,32]. In contrast to apoE in plasma, apoE in the CSF [31,33] and apoE secreted from astrocytes [34] are more heavily glycosylated and sialylated. CSF apoE is five times more glycosylated than plasma apoE [33]. Moreover, *APOE* genotype also affects the percentage of glycosylation in CSF [33]. *APOE2* homozygotes have a higher percentage of glycosylated apoE in the CSF than *APOE4* [33]. The biophysical properties of the protein changes with the addition of sialic groups, namely a reduction of the isoelectric point of apoE [35,36] and an altered solubility in brain tissue [37]. Compared to apoE3, apoE4 is more sialylated, evidenced by a shift in isoelectric point, and more solubility in tris buffered saline buffer [37].

Studies of apoE in the CSF and periphery have provided important information on the effects of *APOE* genotype on apoE production and secretion in the context of AD. Compared to *APOE3*, *APOE4* carriers have reduced levels of apoE, whereas *APOE2* carriers have elevated levels of apoE in the CSF and plasma [38]. This is in agreement with the effects of *APOE* genotype on apoE levels in brain parenchyma (apoE4 < apoE3 < apoE2), and in *APOE* targeted-replacement mice [39,40]. However, other studies show no differences in apoE levels among *APOE* genotypes in the CSF [41] or between AD patients and control subjects [42].

## 5. Effects of apoE Isoforms on Cholesterol Synthesis and Transport/Efflux

In the brain there are two major pools of CNS cholesterol. The first pool, which represents the vast majority of cholesterol (70–90%), is found in the myelin sheaths of oligodendroglia that surrounds axons [43]. Cholesterol synthesis in the brain is highest in oligodendrocytes specially during periods of development [43]. In the mature brain, cholesterol synthesis continues at a lower rate. The second pool of cholesterol derives from plasma membranes of neurons and glia [43]. Astrocytes, which account for up to 50% of all brain cells in humans, provide the bulk of this second pool of cholesterol [44,45].

In astrocytes, cholesterol synthesis is tightly regulated by an internal feedback loop. If intracellular levels of cholesterol are low, an increase in cholesterol synthesis is promoted by proteolysis of sterol regulatory element-binding proteins (SREBPs), which increases cholesterol synthesis and endocytosis [46,47]. When the levels of intracellular cholesterol are high, transcription for cholesterol transport proteins is increased to enhance the efflux of lipids. This process is mediated by the liver X receptors (LXRs) and the retinoid X receptor (RXR) that belong to the type II family of nuclear receptors that undergo obligate heterodimerization [48]. The LXR/RXR heterodimer interacts with sequence-specific DNA elements positioned close to enhancers or promoters of their target genes, including the *ABCA1*, *ABCG1,* and *APOE*, thus acting directly to upregulate their transcription (Figure 1A) [48,49].

ApoE plays an integral role in maintaining lipid homeostasis. Mature neurons have a high demand for cholesterol and while they can synthesize it, under physiological conditions additional apoE-associated cholesterol is required [45,46]. ApoE initiates the formation of high-density lipoprotein (HDL)-like particles by accepting cholesterol and phospholipids through the activity members of the ATP-binding cassette (ABC) family of transporters, ABCA1 and ABCG1 (Figure 1B) [50]. ABC transporters use ATP to translocate a wide variety of substrates across extra- and intracellular membranes. The importance of ABCA1 in AD is discussed in more detail in Section 8.1 below. Less investigated in the context of AD is the ABCG1 transporter. ABCG1 belongs to the G subfamily of the ABC superfamily and is expressed in both neurons and astrocytes [51]. While CSF derived from *ABCG1*
^−^/_−_ mice has increased levels of amyloid β (Aβ)_(1-42)_ compared to wild-type mice [51], overexpressing ABCG1 did not alter the levels of Aβ, plaques, apoE levels, cholesterol efflux, or cognitive performance in mouse models of AD [52]. Thus, the ABCA1 and ABCG1 transporters differ somewhat in their involvement in AD pathogenic processes.

ApoE-containing lipoproteins and lipid complexes interact with cell surface heparin sulfate proteoglycan and cell membrane associated receptors, including the LDL receptor and the LDL receptor-related protein 1 (LRP1) in neurons (Figure 1C). This interaction promotes cellular uptake and redistribution of cholesterol/lipids and storage. Intracellular cholesterol is converted to oxysterols for clearance, a mechanism mediated in the CNS by cholesterol 24S-hydroxylase (CYP46A1) [53]. 24S-hydroxycholesterol can cross the BBB, enter the peripheral circulation and be eliminated as bile from the body [54].

Cholesterol and lipids are distributed to neurons and other brain cells in order to maintain proper cellular function, including neuronal growth, repair and remodeling of membranes, organelle biogenesis, and synaptogenesis [47,55,56,57,58]. Furthermore, CNS repair involves the interaction of apoE with lipid debris and degenerating membranes after injury [59].

## 6. Effects of apoE Isoforms on Lipid Homeostasis

Cholesterol homeostasis is vital for normal brain function, as 25% of all cholesterol in the body is found in the brain. The most recognized function of apoE is transport of cholesterol and other lipids, although apoE is also important in synaptic plasticity, signal transduction and immunomodulation [32,60,61,62]. Peripheral apoE is incorporated into lipoprotein particles in the plasma, and CNS apoE is incorporated into lipoprotein particles in the CSF and in the interstitial fluid of brain parenchyma [30,63].

An imbalance in cholesterol homeostasis is associated with an increased risk of neurodegenerative disorders such as AD, Huntington’s disease (HD), Parkinson’s disease (PD), Niemann-Pick type C disease (NPC), and Smith–Lemli Opitz syndrome (SLOS) [54]. In NPC, neurons sequester unesterified cholesterol and other lipids in late endosomes and/or lysosomes, leading to an unequal distribution of cholesterol between neuronal cell bodies and distal axons. As a result, there is a massive loss of neurons, particularly Purkinje cells in the cerebellum [54]. SLOS is caused by mutations in the gene encoding 7-dehydrocholesterol reductase, the enzyme that catalyzes the final step in the cholesterol biosynthetic pathway, thus causing an elevation of the potentially toxic cholesterol precursor, 7-dehydrocholesterol, and a reduction in cholesterol in cells [54]. In HD, in which striatal and cortical neurons are particularly susceptible to the toxic mutant huntingtin protein, the cholesterol biosynthetic pathway is markedly altered [64]. Expression of mutant huntingtin in neurons downregulate the expression of SREBP-1, the regulator of the cholesterol biosynthetic pathway, thus reducing the levels of cholesterol in the cell [64]. Addition of cholesterol to striatal neurons expressing mutant huntingtin increased their survival, suggesting that the deficit in cholesterol contributes directly to the neurological phenotype of HD [54,64]. Cellular cholesterol accumulation can lead to cytotoxicity, inflammation and alteration of plasma membrane composition [54], whereas cholesterol depletion can lead to synaptic and dendritic spine degeneration, failed neurotransmission, and decreased synaptic plasticity [43].

Due to their structural differences, apoE isoforms exhibit differential abilities of binding/transporting cholesterol and phospholipids. ApoE2 is the most efficient isoform in promoting cholesterol efflux from cells compared to apoE3, and apoE3 is more efficient compared to apoE4 [65,66,67]. *APOE2* homozygotes exhibit defective binding to LDL receptors, and thus these individuals may experience type III hyperlipoproteinemia and premature atherosclerosis, especially in the presence of other conditions such as diabetes, obesity, hypothyroidism or estrogen deficiency. The overproduction of very low-density lipoprotein (VLDL), or the presence of fewer LDL receptors, overwhelms the limited ability of apoE2 to mediate the clearance of triglyceride- and cholesterol-rich β-VLDL in plasma [12].

Cholesterol accumulation in *APOE4* astrocytes is not only due to reduced apoE levels, but also to a dysregulated expression of genes involved in lipid metabolism, such as cholesterol biosynthesis-related genes and to impaired lysosomal cholesterol degradation processes [68,69]. Abnormal cholesterol metabolism by *APOE4* astrocytes would result in altered cholesterol transport to other cell-types and functional deficits in neurons [70]. *APOE4* homozygotes have elevated cholesterol levels in the plasma and increased 24S-hydroxycholesterol in the CSF [71,72]. ApoE4 is associated with smaller lipoproteins [73]. ApoE4-containing lipoproteins promote less cholesterol efflux than apoE3-containing lipoproteins [73]. Moreover, *APOE4* carriers have more lipid-depleted lipoproteins in their CSF than *APOE4*-negative individuals [74,75]. Thus, *APOE4* genotype has profound effects on lipid metabolism, transport and homeostasis.

Intracellular accumulation of lipids in lipid droplets is central to cellular lipid homeostasis, and *APOE* allele-dependent [76,77]. *APOE4* astrocytes accumulate more and in smaller lipid droplets compared to *APOE3* astrocytes [76]. Lipid droplets serve two purposes: First, to sequester free fatty acids that may be cytotoxic in the cytoplasm; and second, become an energy rich storage pool for cellular metabolic needs [76]. In neurons, elevation of ROS results in increased levels of peroxidized lipids, which may cause a disruption in lipid homeostasis, mitochondrial dysfunction and neuronal toxicity. A mechanism used by neurons to mitigate this toxicity is to transfer the burden of lipid accumulation and subsequent clearance to astrocytes [77]. The lipid accumulates inside neighboring astrocytes as lipid droplets, a process that is considered neuroprotective [77].

Additionally, *APOE* genotype is associated with susceptibility to oxidative stress and lipid peroxidation, and it is observed earlier in the progression of AD [78,79]. *APOE4* carriers with AD pathology presented higher levels of oxidative stress and elevated lipid peroxidation in a brain area-specific manner compared to *APOE3* homozygotes [79,80]. Furthermore, synaptosomes isolated from human *APOE4* targeted-replacement mice presented higher levels of reactive oxygen species (ROS) and lipid oxidation compared to *APOE3*, and *APOE3* compared to *APOE2* [81].

ApoE2 and apoE3 bind more lipid peroxidation products than apoE4, a mechanism dependent on the number of cysteine residues in the protein [78,79,82] (Table 1). ApoE keeps these neurotoxic agents from damaging neuronal proteins, but since apoE4 lacks these residues, lipid peroxidation products affect the integrity of neuronal proteins [82]. Together, these results reflect the importance of apoE as a modulator of oxidative stress and lipid peroxidation in cells.

## 7. Effects of apoE Isoforms on Lipidation

ApoE must be secreted and properly lipidated in order to perform its normal functions, such as lipid/cholesterol transport, synapse regeneration, immune modulation, and clearance/degradation of Aβ [83,84,85]. ApoE4 is poorly lipidated and expressed in lower levels compared with apoE2 and apoE3 [83,84,86,87]. The apoE function and receptor binding capacity are dependent on the lipidation status of apoE [88], as key residues involved in receptor binding become unburied when apoE is lipidated [15]. The deficiency of apoE4 lipidation suggests that increasing the lipidation of apoE in general may be a viable therapeutic avenue for AD and other neurological disorders.

Biophysical studies using lipid-free apoE and HDL-like discoidal apoE particles of all three apoE isoforms showed that lipid-free apoE has the tendency to aggregate in vitro in an isoform-dependent manner (apoE4 > apoE3 > apoE2), and lipidation of apoE impedes the formation of aggregates [89]. Lipid-free apoE oligomerizes through a monomer-dimer-tetramer association process [90] but also assembles into higher molecular weight aggregates [91,92]. Higher levels of aggregation of apoE4 might contribute to its toxic effects to neuronal cells.

Lipidated apoE structures vary with apoE isoform. ApoE2 is associated with larger lipoprotein complexes [84]. *APOE2/3* individuals have largest apoE complexes in the CSF compared to *APOE4* heterozygote individuals. Based on nondenaturing gel electrophoresis, the CSF of *APOE2* carriers have a continuum of lipid complexes between 720 and 480 kDa, while *APOE4* carriers have smaller complexes with an average of <500 kDa [86]. ApoE4 forms smaller lipid complexes in both wild-type mouse brains [84,93,94], and mouse brains transfected with viral-expressing different ApoE isoforms [84,95]. Lipidated apoE from human *APOE3* or *APOE4* iPSC-derived astrocytes, assessed by nondenaturing gel electrophoresis, was characterized by large (>669 kDa), medium (440–669 kDa) and small particles (<440 kDa) [96]. While the population of large apoE particles was not affected by genotype, *APOE4* astrocytes secreted less middle size and more smaller size particles compared to *APOE3*, providing more data that apoE4 particles are less lipidated [96].

The major lipids found in apoE-containing CNS particles are free cholesterol, phospholipids and triglycerides [97]. ApoE4 particles are associated with less cholesterol than apoE3 [94,96]. ApoE3 particles contain three times more total cholesterol levels per microgram of apoE protein than apoE4 particles [94]. In contrast, triglycerides are present at higher levels in apoE4-containing particles compared with apoE3 particles [97].

Cholesterol is one of the major components of cellular membranes and acts to make membranes more fluid and to decrease membrane permeability to hydrophilic molecules [98]. Importantly, cholesterol is rich in lipid rafts which are essential domains involved in signal transduction, neuronal cell adhesion, axon guidance and synaptic transmission. Cholesterol is increasingly linked to AD pathology. In cultured neurons, transient membrane cholesterol increase results in Aβ_(1-42)_ overproduction, endosomal enlargement, axonal transport abnormalities, and gene expression changes that are reminiscent of early stages of AD [99]. Replacing membrane cholesterol with stigmasterol, a plant sterol that crosses the BBB, decreased amyloid precursor protein (APP) processing and the production of Aβ peptides in mice [100]. Cholesterol is highly enriched in myelin sheaths and allows the propagation of nerve impulses by saltatory mechanisms. Lower concentrations of phosphatidylinositol and sphingomyelin and a higher concentration of phosphatidylcholine are present in synaptosomal low-density membrane fractions from *APOE4*/*APP* mice compared to *APOE3/APP* mice [101].

This difference in the size and composition of lipoprotein particles associated with *APOE4* genotype could translate into less lipid delivery to cells for membrane synthesis or other cellular functions and less lipid clearance from some locations, compromising neuronal health.

## 8. Recalibrating apoE Functions by Increasing Lipidation

Improving the lipidated state of apoE ameliorates cognitive deficits in the presence or absence of amyloid pathology regardless of *APOE* genotype [73,102,103]. In this section we will discuss possible ways of increasing the lipidation of apoE (Figure 2, Table 2).

### 8.1. Small Molecules that Enhance ABCA1-Mediated apoE4 Lipidation

The ABCA1 transporter transfers cholesterol from cells onto lipid-poor apolipoproteins and regulates apoE lipidation in the brain [88]. Reduced amyloid deposition and improved Aβ clearance have been observed in an AD mouse model after overexpressing ABCA1 transporter [132,133], whereas deletion of the *ABCA1* gene decreases levels of apoE protein [134,135] and increases the deposition of Aβ [88,135,136,137]. Studies of brain tissue, CSF, plasma, and primary astrocyte cultures from ABCA1 ^+^/_+_, ABCA1 ^+^/_−_, and ABCA1 ^−^/_−_ mice showed that deletion of ABCA1 markedly affects metabolism of apoE and cholesterol in the CNS and in nascent lipoprotein particles secreted by cultured astrocytes [132]. Loss-of-function mutations in ABCA1 are associated with low plasma levels of apoE [138]. AD and mild cognitive impaired individuals have 30% less ABCA1-mediated cholesterol efflux capacity toward CSF than healthy individuals; this effect was *APOE* genotype-independent [139]. In addition, lipidation of apoE-containing lipoproteins in the brain may act independently of Aβ metabolism by protecting the integrity of the BBB and maintaining normal cerebrovascular function [140].

One approach to increasing the activation of the ABCA1 transporter is through regulation of micro-RNA expression. Overexpression of micro RNA-33 (miR-33), which is highly expressed throughout the brain and most abundantly in neurons, impairs cellular cholesterol efflux and increases extracellular Aβ levels by promoting Aβ secretion and impairing Aβ clearance [104]. In contrast, deletion of miR-33 in mice increases ABCA1 levels and apoE lipidation, while decreasing endogenous Aβ levels in cortex. Importantly, pharmacological inhibition of miR-33 via antisense oligonucleotide targeted to the brain decreases Aβ levels in cortex of *APP/PS1* mice [104]. Cortex of mir-33-deficient mice, compared with wild-type mice, contains increased levels of ABCA1 protein and larger apoE-containing lipoproteins, suggesting that miR-33 plays a dominant role in regulating ABCA1 expression and apoE lipidation under normal physiological conditions [104].

A second approach is through the use of small peptides based on the C-terminal domain of apoE as ABCA1 agonists with therapeutic potential [106]. The peptide CS-6253 enhanced lipid efflux through ABCA1 allowing the generation of apoE3-like lipoproteins in young *APOE4*-targeted replacement mice [106]. CS-6253 ameliorated *APOE4*-driven cognitive and brain pathologies through activation of ABCA1 [73]. This effect was associated with a reversal of the apoE4-driven Aβ_(1-42)_ accumulation and tau hyperphosphorylation in hippocampal neurons, as well as of the synaptic impairments and cognitive deficits associated with *APOE4* phenotype [107].

A second peptide with therapeutic potential is Ac-hE18A-NH2, a hybrid apoE/apoA-I-mimetic peptide. Ac-hE18A-NH2 ameliorated the inhibitory effects of Aβ_(1-42)_ on secretion of apoE in U251 astrocyte cell line [108]. In vivo administration of Ac-hE18A-NH2 to *APP/PS1ΔE9* mice for six weeks improved cognition, decreased amyloid plaque deposition, reduced activated microglia and astrocytes, and increased CNS apoE levels [108].

Another small peptide is 4F, which consists of 18 amino acids containing four phenylalanine residues that mimic HDL function. 4F induces a selective and robust concentration- and time-dependent increase in apoE secretion and lipidation in primary human astrocytes and in primary murine glial cells, without causing cell death [109]. These observed effects were dependent on the presence of the ABCA1 transporter. Co-treatment with 4F counteracts the inhibitory effects of Aβ on both apoE secretion and lipidation in primary mouse and human astrocytes [109]. Furthermore, 4F has been tested in three human clinical trials for cardiovascular disease: 4F improved HDL anti-inflammatory properties, and was found to be safe and well-tolerated when administered orally or by injections [110,111].

ApoE lipidation could also be affected by other molecules that alter ABCA1 activity. ABCA1 recycling and degradation is regulated by ADP-ribosylation factor 6 (ARF6). ApoE4 promotes greater expression of ARF6 compared with apoE3, trapping ABCA1 in late-endosomes and impairing its recycling to the cell membrane, thus lowering ABCA1-mediated cholesterol efflux activity. Reduced ABCA1 activity increases the prevalence of lipid-free apoE particles, and lowers Aβ degradation capacity [105]. Thus, regulating the expression of ARF6 with antisense oligonucleotides in a way that favors the recycling of ABCA1 may also be beneficial for *APOE4* carriers.

### 8.2. Liver X Receptor (LXR) and Retinoid X Receptor (RXR) Agonists

Nuclear receptor agonists upregulate *ABCA1*, *ABCG1,* and *APOE* gene expression, which increases apoE lipidation, facilitates Aβ clearance, reduces amyloid pathology, and reverses memory deficits in an amyloid mouse model [49,103,116]. Bexarotene and LG100268 are RXR agonists that promote dimerization and activation of both LXRs and peroxisome proliferator-activated receptor-gamma (PPAR-γ), resulting in the increase of apoE lipidation and ABCA1 expression in AD animal models [115,116,117]. In vitro and in vivo studies show that bexarotene suppresses inflammation and astrogliosis, activates microglial phagocytosis, and improves neuronal survival, differentiation and neuronal projections [118,119]. The effects of bexarotene on soluble Aβ_(1-40)_ and Aβ_(1-42)_ levels and plaque deposition are dependent on the presence of both *APOE* and *ABCA1* [112,115].

In addition, two prenylated flavanoids isolated from the root of *Sophora tonkinensis* (SPF1 and SPF2) have selective RXR agonist activity and promote ABCA1 upregulation, thus promoting apoE lipidation [120,121]. They protect differentiated PC12 cells from Aβ-induced neurotoxicity. When given in combination with the LXR agonist T0901317, these effects are markedly enhanced [120]. SPF1 and SPF2 also suppresses the cytokine production of IL-1β, IL-6, and TNF-α in vitro [121].

LXR activation by T0901317 or GW3965 altered Aβ production in neuronal and non-neuronal cells in vitro and in vivo [114,141]. Short-term administration of T0901317 or GW3965 improved fear conditioning behavioral outcomes, increased apoE lipidation, and promoted the association of microglia with Aβ plaques. Long-term administration treatment restored object recognition and reduced plaque formation by 50% in mouse models of AD [112,113,114].

Although nuclear receptor agonists may be promising therapeutic targets for AD, they produce unwanted peripheral effects on triglyceride production and liver health [142,143,144]. LXR activation induces the expression of genes associated with lipogenesis, a mechanism mediated by the induction of transcription factors such as the carbohydrate-response element-binding protein (ChREBP), PPAR-γ, and the SREBP-1 [143,144]. Moreover, LXRs regulate enzymes involved in the biosynthesis of saturated or monounsaturated fatty acids (e.g., acetyl-CoA carboxylase, fatty acid synthase, steroyl-CoA desaturase), thus increasing the levels of triglyceride and phospholipid in the liver in different animal models, leading to hepatic steatosis, stimulation of VLDL secretion and elevated plasma triglyceride levels [144]. Hepatic steatosis was also observed in human clinical trials, limiting the therapeutic development of synthetic LXR agonists [144]. It is possible that CNS-specific LXR activation could be useful while avoiding side effects in the periphery.

### 8.3. Small Molecules as apoE4 Structure Correctors

Structure correctors could prevent the apoE4-associated neuropathology by changing the conformation of apoE4 into an apoE3-like structure, thus allowing proper lipidation of the apoE protein and greater transport of phospholipids and cholesterol [12]. As mentioned in Section 7, apoE4 is more easily self-aggregated, thus reducing its lipidation capacity. The use of apoE4 structure correctors would reduce apoE aggregation and facilitate apoE lipidation, becoming an appealing therapeutic potential for AD treatment. ApoE4 is also targeted for degradation more readily than apoE3, perhaps due to its altered folding [39]. ApoE4 is recognized by neuron-specific proteases generating neurotoxic C-terminal fragments [145]; structure correctors could diminish the production of these neurotoxic fragments.

A cellular fluorescence resonance energy transfer (FRET) assay has been used for the identification of small-molecule structure correctors to identify molecules that block the ionic interaction between Arg-61 and Glu-255. ApoE4 labeled at the N-terminus with green fluorescent protein and the C-terminus with Escherichia coli dihydrofolate reductase [122,146] allowed identification of small-molecule structure correctors to prevent FRET emission signal that occurs when the N-terminal and the C-terminal of apoE4 are in close proximity. The experimental compound PH002 was identified that dose-dependently mitigated the toxic effect of apoE4 in neurons derived from human iPSCs by decreasing the levels of apoE4 fragments, increasing GABAergic neuron numbers and GAD67 levels, reducing hyperphosphorylated tau levels, and decreasing the levels of Aβ_(1-40)_ and Aβ_(1-42)_ production and/or secretion [122]. Two other compounds, GIND-25 and GIND105, were also designed to restore apoE4 to an apoE3-like protein with respect to its lipid-binding characteristics and its effects on cultured neurons [123,124,125].

### 8.4. Anti-apoE4 Immunotherapy

Introduction or generation of antibodies against the non-lipidated form of apoE4 could reduce toxic effects of non-lipidated apoE4. Non-lipidated apoE4 protein has a greater tendency to aggregate than non-lipidated apoE3 [126], and lipidating apoE *ex vivo* reduces apoE aggregation [89]. Targeting a specific conformational form of non-lipidated apoE reduces Aβ pathology in mouse models of AD [127]. The anti-human apoE antibody HAE-4 (HAE-4), that specifically recognizes human apoE4 and apoE3, preferentially binds non-lipidated as compared to lipidated apoE [127]. When delivered centrally (injection or adeno-associated virus) or peripherally, HAE-4 preferentially binds to apoE aggregates, and reduces Aβ deposition and Aβ accumulation in the brain of *APP/PS1-21/APOE4* mice without altering levels of total apoE in plasma, CSF, or the extracellular space of the brain [127]. Antibodies mediating a protective effect by targeting aggregated or non-lipidated apoE in plaques and inciting microglial response, leading to clearance, is a promising approach for treating AD.

### 8.5. Recalibrating apoE Function by Using AAV-APOE2 Biologic Therapy

Individuals homozygotes for the *APOE4* allele have the highest risk of developing AD. On the other hand, the presence of the *APOE2* allele, markedly reduces the risk of developing late-onset AD. Based on this genetic data, it was hypothesized that expressing the protective *APOE2* allele by genetically modifying the CNS of *APOE4* homozygotes, converting them into *APOE2/4* heterozygotes, could reverse or prevent the progression of the disease. Since apoE2 is associated with larger lipoproteins than apoE3 and apoE4 (Section 7, “Effects of apoE isoforms on lipidation”), this approach would introduce more lipidated forms of apoE to at-risk individuals. AAV delivery of *APOE2* into the CSF of the lateral ventricle of mice overexpressing mutant *APP* or *APP/PS1/APOE4* transgenes showed protective effect against the pathology associated apoE4 expression [128,129]. Nondenaturing gel electrophoresis demonstrated that *APOE2* expressed by the AAV vector was lipidated [130], as previously reported for endogenous apoE [131]. Moreover, three routes of AAV delivery (intraparenchymal, intraventricular, and intracisternal) of the *APOE2* gene to the CNS in non-human primates were compared and showed to be both efficacious and safe, resulting in therapeutic levels of apoE2 in the CSF (>1 μg/mL) [128] (human and rodent apoE levels in the CSF are 1–5 μg/mL [147]). Lastly, intracerebroventricular injection of AAV vectors expressing *APOE2* in the *APOE4* transgenic mice enhanced apoE lipidation, increased levels of apoE-associated cholesterol, and decreased endogenous Aβ [84]. Conversely, apoE4 overexpression in *APOE4* transgenic mice increased poorly-lipidated apoE lipoprotein particles and levels of endogenous Aβ, and decreased levels of apoE-associated cholesterol [84].

## 9. Concluding Remarks

ApoE is the main lipid carrier in the CNS and the current evidence highlights the importance of *APOE* isoforms in modulating the pathogenesis of AD. ApoE isoforms not only affect Aβ clearance and aggregation, but also neuronal growth, repair and remodeling of membranes, synaptogenesis, and neuroinflammation. Here we highlight therapeutic approaches targeted to *APOE* lipidation, which have the potential to correct or prevent several outcomes associated with neurodegeneration. Many of these approaches have been validated using diverse cellular and animal models. *APOE*-targeted therapies may be more effective at preventing AD rather than treating existing AD and neurodegeneration. The benefits of increasing lipidation and decreasing lipid-free availability might be more beneficial with *APOE4* carriers, which account for large percentages of control and AD populations. Current genotyping technologies and platforms promote access to information about the predisposition of an individual to AD at very early ages, thus allowing preventive treatments for high risk populations throughout life. These *APOE*-targeted therapies need also to take into account effects of altering apoE and lipid homeostasis in the periphery. The data discussed herein point towards a therapeutic model in which increasing lipidation, while simultaneously decreasing lipid-free apoE by a combined therapy, would be an appealing approach to ameliorate or prevent AD.

## Figures and Tables

**Figure 1 ijms-21-06336-f001:**
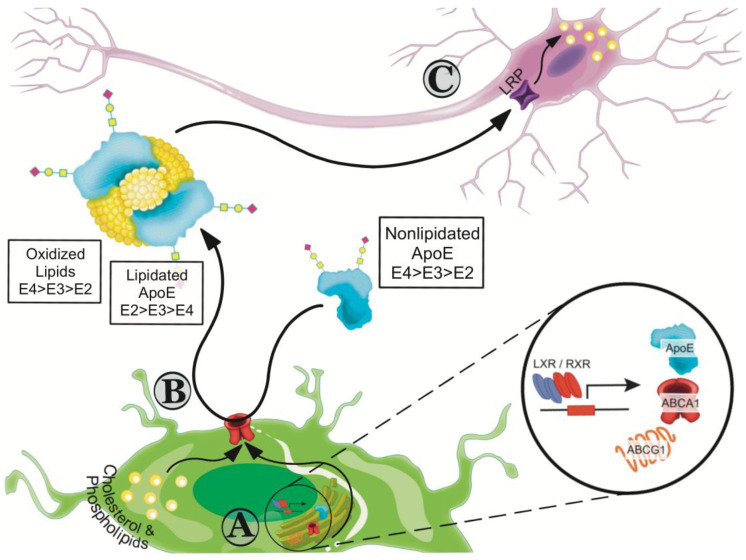
apoE and cholesterol transport and efflux. (**A**) In astrocytes, cholesterol synthesis is regulated by the liver X receptors (LXRs) and the retinoid X receptor (RXR). The LXR/RXR heterodimer interacts with sequence-specific DNA elements positioned close to enhancers or promoters of their target genes, including the *ABCA1*, *ABCG1,* and *APOE*, thus acting directly to upregulate their transcription. (**B**) apoE initiates the formation of high-density lipoprotein (HDL)-like particles by accepting cholesterol and phospholipids through the ABCA1 and ABCG1 transporters. (**C**) apoE-containing lipoproteins and lipid complexes interact with cell surface heparin sulfate proteoglycans and cell membrane associated receptors, including the LDL receptor and the LDL receptor-related protein 1 (LRP) in neurons. This interaction promotes cellular uptake and redistribution of cholesterol to maintain proper cellular function, including neuronal growth, repair and remodeling of membranes, organelle biogenesis, and synaptogenesis.

**Figure 2 ijms-21-06336-f002:**
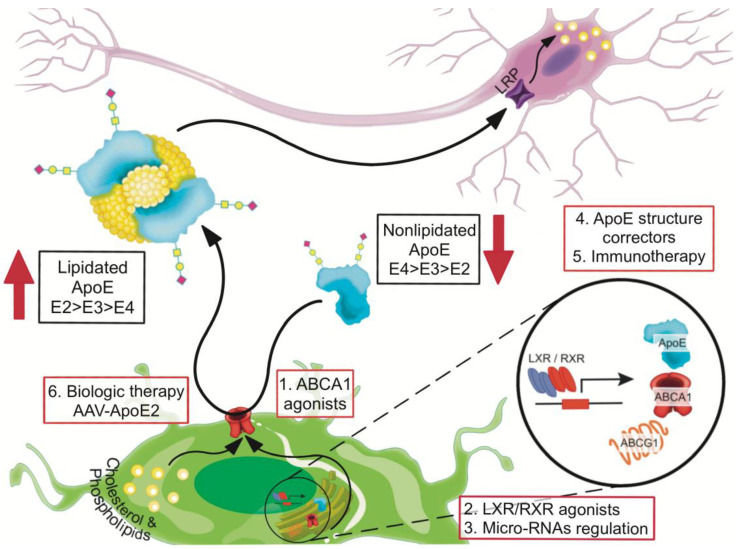
Recalibrating apoE functions by increasing lipidation. Therapeutic strategies to improve the lipidated state of apoE include the use of: (1) Small molecules that enhance ABCA1 activity, (2) Liver X receptor (LXR) and retinoid X receptor (RXR) agonists to increase the expression of *ABCA1*, *ABCG1* and *APOE*, (3) regulation of micro-RNAs to increase ABCA1 expression, (4) small molecules as apoE4 structure correctors, (5) anti-apoE4 immunotherapy targeting non-lipidated apoE, and (6) recalibrating apoE function by using AAV-*APOE2* biologic therapy.

**Table 1 ijms-21-06336-t001:** Structural differences among apolipoprotein E (apoE) isoforms.

Isoform	Amino Acids (112, 158)	Structural Description
ApoE2	Cys, Cys	Cys-158 at the N-terminal domain reduces receptor binding. Arg-61 is internal to the helical domain of the N-terminus. Ability to dimerize through cysteine-cysteine bonds, forming homodimers and multimers.
ApoE3	Cys, Arg	Presence of salt bridge between Arg-158 and Asp–154. Arg-61 is internal to the helical domain of the N-terminus. Ability to dimerize through a cysteine-cysteine bond, forming homodimers.
ApoE4	Arg, Arg	Asp-154 interacts with Arg-150, altering the receptor binding region. Ionic interaction between Arg-61 and Glu-255 in the C-terminal domain. No cysteine residue.

**Table 2 ijms-21-06336-t002:** Therapeutic approaches to recalibrate apoE functions by increasing lipidation.

Class	Description	Example	References
ABCA1 agonist	Antisense oligonucleotides	miR-33 ARF6	[104] [105]
	Small peptides	CS-6253 Ac-hE18A-NH2 4F	[73], [106,107] [108] [73], [109,110,111]
Nuclear Receptor agonist	LXR agonist	TO901317 GW3965	[112,113,114] [112,113,114]
	RXR agonist	Bexarotene LG100268 SPF1 and SPF2	[115,116,117,118,119] [115,116,117] [120,121]
Structure corrector	Small molecule that corrects apoE4 structure	PH002 GIND105 and GIND-25	[122] [123,124,125]
Immunotherapy	Targets non-lipidated apoE4	HAE-1 and HAE-4	[126,127]
Biologics	AAV-directed therapy	AAV-expressing human *APOE2* gene	[84], [128,129,130,131]

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
