# Peer review of "ApoE Lipidation as a Therapeutic Target in Alzheimer’s Disease"

_ijms, 2020, doi:10.3390/ijms21176336_

Round 1

Reviewer 1 Report

This review focuses  on apoE , the major cholesterol carrier in the brain, and on the mechanisms underlying the pathological effects of its allelic variant APOE4, which is the most prevalent genetic risk factor of Alzheimer’s disease (AD)  .The first part of this paper includes a comprehensive review of the structure and  function of the different apoE isoforms and of role of apoE lipidation in mediating the pathological effects of apoE4.

The second  part of the paper focuses on apoE lipidation as a therapeutic target in  AD . This presents a novel and emerging approach to the treatment of AD  which focuses on reversal of the hypolipidation of apoE4.  This is discussed in  a clear and concise way and , to our knowledge is  the first review which focuses on  this very promising therapeutic approach to AD.  While there is a general agreement in the field that apoE4 is a promising therapeutic target .The field lacks coherent suggestions , such as that presented in this paper ,as to how this problem could be tackled effectively. 

Author Response

Reviewer 1:

This review focuses on apoE, the major cholesterol carrier in the brain, and on the mechanisms underlying the pathological effects of its allelic variant APOE4, which is the most prevalent genetic risk factor of Alzheimer’s disease (AD). The first part of this paper includes a comprehensive review of the structure and  function of the different apoE isoforms and of role of apoE lipidation in mediating the pathological effects of apoE4.

The second  part of the paper focuses on apoE lipidation as a therapeutic target in  AD. This presents a novel and emerging approach to the treatment of AD  which focuses on reversal of the hypolipidation of apoE4.  This is discussed in  a clear and concise way and, to our knowledge is  the first review which focuses on  this very promising therapeutic approach to AD.  While there is a general agreement in the field that apoE4 is a promising therapeutic target .The field lacks coherent suggestions , such as that presented in this paper ,as to how this problem could be tackled effectively. 

Response: We appreciate the reviewer’s enthusiasm and do hope that our review serves a useful purpose in the literature.

Reviewer 2 Report

This manuscript by Lanfranco, Ng, and Rebeck is a comprehensive review of the current literature surrounding ApoE lipidaiton and its use as a therapeutic target in Alzheimer’s disease. This review provides an excellent summary of the scientific literature focused on ApoE’s role in lipid homeostasis, the effects of APOE genotype on ApoE’s function, how APOE genotype affects lipid trafficking, and ways to increase lipidation of ApoE to decrease CNS impairments. It focuses on a critical subject and is well written. While minor, there are several grammatical errors and stylistic edits suggested (as outlined below), and several instances in which sentence clarity should be improved and/or additional text would be helpful.

Section 2 (Introduction):

The first sentence of paragraph 2 is a run on sentence making it long and difficult to follow. It should be split into two sentences.

In the second paragraph, line 40, “The alleles combinations” should read “The allele combinations” or “The allelic combinations”

Section 3 (Structures and functions of apoE isoforms):

Paragraph 1, sentence 2, line 49: The sentence would read better with minor rearrangement by removing “which” after the comma and inserting an “and” between protein and harbors later in the sentence.

Paragraph 2, sentence 4, line 59:  Please revise this sentence to make it more cohesive and easier understand how lipid binding affects the conformation of apoE.  It would be best to split the sentence into two sentences on line 63 following, “circumscribing the edge of the nano disc [17-19].” This way, it is easier to differentiate between the differences in conformation changes in the N and C termini of apoE.

Paragraph 4, sentence 3, line 76: This sentence should be split into two separate sentences. It should read “In apoE4, Asp-154 interacts with Arg150 altering the receptor binding region. As a consequence of this new interaction, the side chain…”

Section 4 (CNS apoE protein):

Paragraph 2, sentence 2, line 89: The sentence is a bit confusing. It is the reviewer’s impression that apoE from both astrocytes and CSF are more heavily glycosylated and sialylated than apoE from the plasma. However, the text makes it difficult to decipher.

Section 5 (ApoE and cholesterol synthesis and transport/efflux):

Paragraph 1, sentence 3, line 107: It us unclear whether the author meant to use “specially” to describe a unique increase in cholesterol synthesis during development, or if they meant to say “especially” to denote that oligodendrocytes synthesize more cholesterol during development.

Paragraph 1, sentence 4 and 5, lines 107-109: The paragraph would be more cohesive if these sentences were rearranged, perhaps even just switched so the paragraph reads: “Cholesterol synthesis continues in the mature brain at a lower rate. The other pool of cholesterol is derived from neuronal and glial plasma membranes [43].” It would make the paragraph less choppy and help to connect the ideas better.

Paragraph 2, sentence 3, line 114: The sentence is unclear and hard to follow without an active verb. Sentence should read: “When the levels of intracellular cholesterol are high, transcription of cholesterol transport proteins is increased to increase the efflux of lipids.”

Section 6:

Paragraph 3, sentence 2, line 184: The authors should split the sentence into separate sentences to avoid a run-on sentence. The addition of a period after “phospholipids” on line 84 and capitalization of “ApoE2 is the most…” would fix the run on sentence.

Paragraph 4, sentence 4, line 196: The authors should consider splitting the sentence into separate sentences by adding a period after “smaller lipoproteins” on line 197.

Section 7:

Paragraph 2, sentence 2, line 213: The sentence is mostly clear, but has awkward wording at the end. The review interprets it as lipid-free apoE oligomerizing through a specific association process to assemble a tetramer with a higher molecular weight.

Paragraph 3, sentence 3, line 220: The author’s should revise the wording at the end of the sentence. The reviewer took away that ApoE4 formed smaller lipid complexes in both normal mouse brains and mouse brains transfected with viruses expressing different ApoE isoforms, but the wording made it confusing and unclear.

Paragraph 4, sentence 2, line 228: The reviewer suggests splitting the sentence to make it easier to follow and differentiate between cholesterol association in the two different isoforms. Add a period after “less cholesterol than apoE3 [87,89].”

Section 8.3

The reviewer would like to see more elaboration on why this would be a significant or beneficial therapeutic approach. The summary of the study and its findings are great, but why is it significant? How does it tie into the rest of the review?

Author Response

This manuscript by Lanfranco, Ng, and Rebeck is a comprehensive review of the current literature surrounding ApoE lipidaiton and its use as a therapeutic target in Alzheimer’s disease. This review provides an excellent summary of the scientific literature focused on ApoE’s role in lipid homeostasis, the effects of APOE genotype on ApoE’s function, how APOE genotype affects lipid trafficking, and ways to increase lipidation of ApoE to decrease CNS impairments. It focuses on a critical subject and is well written. While minor, there are several grammatical errors and stylistic edits suggested (as outlined below), and several instances in which sentence clarity should be improved and/or additional text would be helpful.

Section 2 (Introduction):

The first sentence of paragraph 2 is a run on sentence making it long and difficult to follow. It should be split into two sentences.

Response: We have split the previous sentence in two to improve the clarity of the text (line 38).

In the second paragraph, line 40, “The alleles combinations” should read “The allele combinations” or “The allelic combinations”.

Response: We changed “alleles combinations” to “allelic combinations” (line 44).

Section 3 (Structures and functions of apoE isoforms):

Paragraph 1, sentence 2, line 49: The sentence would read better with minor rearrangement by removing “which” after the comma and inserting an “and” between protein and harbors later in the sentence.

Response: We rearranged the sentence to read “The C-terminal domain (residues 206–299), that harbors the major lipid binding region, presents a more relaxed structure with α-helices generating a largely exposed hydrophobic surface” (line 56).

Paragraph 2, sentence 4, line 59:  Please revise this sentence to make it more cohesive and easier understand how lipid binding affects the conformation of apoE. It would be best to split the sentence into two sentences on line 63 following, “circumscribing the edge of the nano disc [17-19].” This way, it is easier to differentiate between the differences in conformation changes in the N and C termini of apoE.

Response: We rearranged and split the original sentence to read “Lipid binding reorients the α-helices of the C-terminal domain of apoE perpendicularly to the acyl chains of the lipids, circumscribing the edge of the nanodisks [17-19]. However, the conformational change on the N-terminal domain has not converged towards a single model” (line 66).

Paragraph 4, sentence 3, line 76: This sentence should be split into two separate sentences. It should read “In apoE4, Asp-154 interacts with Arg150 altering the receptor binding region. As a consequence of this new interaction, the side chain…”

Response: We have split the previous sentence in two to improve the clarity of the text (line 80).

Section 4 (CNS apoE protein):

Paragraph 2, sentence 2, line 89: The sentence is a bit confusing. It is the reviewer’s impression that apoE from both astrocytes and CSF are more heavily glycosylated and sialylated than apoE from the plasma. However, the text makes it difficult to decipher.

Response: rearranged the original sentence to read “In contrast to apoE in plasma, apoE in the CSF [31, 33] and apoE secreted from astrocytes [34] are more heavily glycosylated and sialylated” (line 98).

Section 5 (ApoE and cholesterol synthesis and transport/efflux):

Paragraph 1, sentence 3, line 107: It us unclear whether the author meant to use “specially” to describe a unique increase in cholesterol synthesis during development, or if they meant to say “especially” to denote that oligodendrocytes synthesize more cholesterol during development.

Response: rearranged the original sentence to read “The first pool, which represents the vast majority of cholesterol (70-90%), is found in the myelin sheaths of oligodendroglia that surrounds axons [43]. Cholesterol synthesis in the brain is highest in oligodendrocytes specially during periods of development [43] (line 115).

Paragraph 1, sentence 4 and 5, lines 107-109: The paragraph would be more cohesive if these sentences were rearranged, perhaps even just switched so the paragraph reads: “Cholesterol synthesis continues in the mature brain at a lower rate. The other pool of cholesterol is derived from neuronal and glial plasma membranes [43].” It would make the paragraph less choppy and help to connect the ideas better.

Response: We rearranged the original sentence to read “The first pool, which represents the vast majority of cholesterol (70-90%), is found in the myelin sheaths of oligodendroglia that surrounds axons [43]. Cholesterol synthesis in the brain is highest in oligodendrocytes specially during periods of development [43]. In the mature brain, cholesterol synthesis continues at a lower rate. The second pool of cholesterol derives from plasma membranes of neurons and glia [43]. Astrocytes, which account for up to 40% of all brain cells in humans, provide the bulk of this second pool of cholesterol [44, 45] (line 115).

Paragraph 2, sentence 3, line 114: The sentence is unclear and hard to follow without an active verb. Sentence should read: “When the levels of intracellular cholesterol are high, transcription of cholesterol transport proteins is increased to increase the efflux of lipids.”

Response: We followed the reviewer’s suggestions and the original sentence was replaced with “When the levels of intracellular cholesterol are high, transcription for cholesterol transport proteins is increased to enhance the efflux of lipids” (line 125).

Section 6:

Paragraph 3, sentence 2, line 184: The authors should split the sentence into separate sentences to avoid a run-on sentence. The addition of a period after “phospholipids” on line 84 and capitalization of “ApoE2 is the most…” would fix the run on sentence.

Response: We followed the reviewer’s suggestions and split the sentence in two. A period was added after phospholipids (line 198).

Paragraph 4, sentence 4, line 196: The authors should consider splitting the sentence into separate sentences by adding a period after “smaller lipoproteins” on line 197.

Response: We followed the reviewer’s suggestions and split the sentence in two. A period was added after “smaller lipoproteins” (line 211).

Section 7:

Paragraph 2, sentence 2, line 213: The sentence is mostly clear, but has awkward wording at the end. The review interprets it as lipid-free apoE oligomerizing through a specific association process to assemble a tetramer with a higher molecular weight.

Response: We followed the reviewer’s suggestions and the original sentence was replaced with “Lipid-free apoE oligomerizes through a monomer-dimer-tetramer association process [83] but also assembles into higher molecular weight aggregates [84, 85].” (line 249).

Paragraph 3, sentence 3, line 220: The author’s should revise the wording at the end of the sentence. The reviewer took away that ApoE4 formed smaller lipid complexes in both normal mouse brains and mouse brains transfected with viruses expressing different ApoE isoforms, but the wording made it confusing and unclear.

Response: We thank the reviewer for pointing this out. We clarified the sentence and changed it to “ApoE4 forms smaller lipid complexes in both wild-type mouse brains [77, 86, 87], and mouse brains transfected with viral-expressing different ApoE isoforms [77, 88]” (line 256).

Paragraph 4, sentence 2, line 228: The reviewer suggests splitting the sentence to make it easier to follow and differentiate between cholesterol association in the two different isoforms. Add a period after “less cholesterol than apoE3 [87,89].”

Response: We followed the reviewer’s suggestions and split the sentence in two. A period was added after “less cholesterol than apoE3” (line 264).

Section 8.3

The reviewer would like to see more elaboration on why this would be a significant or beneficial therapeutic approach. The summary of the study and its findings are great, but why is it significant? How does it tie into the rest of the review?

Response: We thank the reviewer for pointing this out. We added at the beginning of section 8.3 a paragraph illustrating the benefits and significance of using apoE correctors as a therapeutic option for AD (line 388).

Reviewer 3 Report

The review is well written and very informative for people interested in the link between ApoE and AD. In my opinion, the work could gain value if the authors address the following minor issues:

  • Make a figure and/or a table describing synthetically section 3, Structures and functions of apoE isoforms, in order to help the reader to focus on the differences between apoE isoforms;
  • Make an informative table listing all the possible compounds (peptides, apoE structure correctors, etc.) which could be used for therapeutic purposes.
  • Mention and discuss the link between apoE and oxidative stress/lipid peroxidation

Author Response

The review is well written and very informative for people interested in the link between ApoE and AD. In my opinion, the work could gain value if the authors address the following minor issues:

Make a figure and/or a table describing synthetically section 3, Structures and functions of apoE isoforms, in order to help the reader to focus on the differences between apoE isoforms;

Response: We agree that this information will be useful for understanding apoE isoform differences. We have now included a table briefly summarizing the differences in structure and function of apoE isoforms (Table 1, line 88).

Make an informative table listing all the possible compounds (peptides, apoE structure correctors, etc.) which could be used for therapeutic purposes.

Response: We agree that this table too would provide a convenient way for reviewers to gain perspective on the therapeutic approaches. We followed the reviewer’s suggestions and included a table summarizing the differences in potential treatments to increase apoE lipidation and the compounds currently available (Table 2, line 299).

Mention and discuss the link between apoE and oxidative stress/lipid peroxidation

Response: We thank the reviewer for asking us to include this interesting area related to apoE isoform differences. We added three paragraphs at the end of section 6 on the effects of apoE isoforms and oxidative stress and lipid peroxidation (line 217). In addition, we added a recent publication about the role of apoE in lipid droplet accumulation to that paragraph (line 217).